# Effects of Eggshell Thickness, Calcium Content, and Number of Pores in Erosion Craters on Hatching Rate of Chinese Alligator Eggs

**DOI:** 10.3390/ani13081405

**Published:** 2023-04-19

**Authors:** Naijing Zhang, Huabin Zhang, Guangwei Fan, Ke Sun, Qingqing Jiang, Zhuowen Lv, Boyang Han, Zhenyuan Nie, Yujie Shao, Yongkang Zhou, Baowei Zhang, Xiaobing Wu, Tao Pan

**Affiliations:** 1College of Life Sciences, Anhui Normal University, Wuhu 241000, China; 2Anhui Provincial Key Laboratory of the Conservation and Exploitation of Biological Resources, College of Life Sciences, Anhui Normal University, Wuhu 241000, China; 3Anhui Chinese Alligator National Nature Reserve, Xuancheng 242099, China; 4School of Life Sciences, Anhui University, Hefei 230039, China

**Keywords:** eggshell, microstructure, thickness, number of pores in erosion craters, hatching rate

## Abstract

**Simple Summary:**

Any abnormalities in the physical properties of an egg can decrease the hatching rate. The embryos of oviparous reptiles obtain Ca from the eggshell. This motivated us to analyze the microstructure and Ca in the eggshells of Chinese alligators. We found that the shells of the eggs with high hatching rates were thicker than those of the eggs with low hatching rates. There were also fewer erosion-crater pores on the surfaces of the eggs with high hatching rates than on the surfaces of the eggs with low hatching rates. Moreover, the shells’ Ca contents were significantly higher in the eggs with high hatching rates than in the eggs with low hatching rates. Cluster modeling indicated that the highest hatching rate occurred when the eggshell thickness was 200–380 µm and there were 1–12 pores. These results suggest that eggs with adequate Ca contents, thicker shells, and less air permeability are more likely to hatch. Furthermore, our findings can inform future studies.

**Abstract:**

The Chinese alligator (*Alligator sinensis*), found only in a small region in southeastern Anhui Province, is listed as critically endangered (CR) by the International Union for Conservation of Nature (IUCN) due to its current declining population trend. Any abnormalities in the physical properties of an egg can decrease the hatching rate. In particular, eggshells play an essential role in embryo development, motivating us to analyze the microstructures of the eggshells of Chinese alligators. In this study, we categorized the eggshells into two groups, based on the hatching rates, and analyzed the relationship between the eggshell parameters (eggshell thickness, calcium content, and number of pores in erosion craters) and the hatching rate, as well as the relationships between the eggshell parameters. We found that the shells of the eggs with high hatching rates were thicker than those of the eggs with low hatching rates. There were also fewer erosion-crater pores on the surfaces of the eggs with high hatching rates than on the surfaces of the eggs with low hatching rates. Moreover, the shell Ca content was significantly higher in the eggs with high hatching rates than in the eggs with low hatching rates. Cluster modeling indicated that the highest hatching rate occurred when the eggshell thickness was 200–380 µm and there were 1–12 pores. These results suggest that eggs with adequate Ca contents, thicker shells, and less air permeability are more likely to hatch. Furthermore, our findings can inform future studies, which will be vital for the survival of the critically endangered Chinese alligator species.

## 1. Introduction

Reproductive success is critical to species survival [1]. In oviparous species, the physical properties of eggs play an essential role in embryo development and successful hatching [2]. Any abnormalities in an egg’s physical characteristics can lead to a decrease in the likelihood of its hatching [1]. Eggshell parameters (e.g., porosity and number of pores) are essential factors influencing the successful hatching of eggs [3,4,5]. The eggshell represents a trade-off between two important functions during embryo development: it must be thick and strong enough to protect the embryo from external damage but not so thick that it is an obstacle to hatching. Furthermore, the eggshell must have sufficient pores to provide oxygen to the developing embryo, but a large number of pores allows pathogenic microorganisms to invade the egg [3].

As mentioned above, the eggshell thickness significantly affects the hatching rate [3]. The thickness of shells substantially differs among hatched, unhatched, and unfertilized eggs [6]. Specifically, eggs that fail to hatch have significantly thinner eggshells than those that successfully hatch. Moreover, the hatching rate of thin-shelled eggs is 3% to 9% lower than that of thick-shelled eggs. Most researchers have reported that thick eggshells provide the following advantages: (1) the full use of nutrients by the embryo (in chicken eggs) [7], (2) a reduced likelihood of bacteria entering the egg [8], (3) a reduced likelihood of egg dehydration [9], and (4) better protection against mechanical damage [10].

The eggshell buffers against changes in the external environment by allowing the exchange of water and gases essential to embryonic development [11,12]. Most squamates lay eggs with leathery shells with high breathability [13]; indeed, the fiber layers of snake eggs, which have leathery shells, is reduced during incubation to increase breathability for the embryo [14]. In contrast, most turtles and all crocodiles lay hard-shelled eggs with very low breathability. To compensate for this low breathability, the eggs harden and become opaque during the second half of incubation. During this period, the embryos rapidly develop, and oxygen consumption and carbon-dioxide production increase exponentially [15,16]. Once hardened (i.e., when they become opaque), the shells of turtles and crocodiles have lower water contents. Removing water from porous eggshells significantly improves their permeability to oxygen and carbon-dioxide gas, thereby promoting the diffusion and exchange of oxygen and carbon dioxide [17]. Conversely, hypoxia reduces the heart rate of American alligators [18] and some birds [19], and it can lead to a decline in both embryo growth and survival in reptiles (turtles and crocodiles), as well as birds [20,21].

The maternal consumption of micronutrients, macronutrients, and fatty acids may influence the hatching rate of alligator eggs [22]. Inadequate provision or the pathological metabolism of calcium (Ca) are common in captive reptiles [23]. Crocodile hatchlings are prone to metabolic bone disease caused by inadequate Ca, of which the main symptoms are kyphoscoliosis and jaw softening (“rubber jaw”), accompanied by changes in tooth structure (“glassy teeth”) [24]; alligator hatchlings fed a Ca-deficient diet are also prone to long-bone and spinal compression fractures [25]. Importantly, the embryos of birds and oviparous reptiles obtain Ca from the eggshell [26].

In this study, we categorized eggshells into two groups (low hatching rate vs. high hatching rate) based on their hatching rate and analyzed the relationship between the eggshell parameters (eggshell thickness, calcium content, and number of pores in erosion craters) and the hatching rate, as well as the relationships between the eggshell parameters. We expected that our results would provide important insights into the factors determining the hatching of alligator eggs, as well as the supplemental feeding of female alligators, and thus improve the hatching rate of Chinese alligator eggs.

## 2. Materials and Methods

### 2.1. Materials

The eggshells of Chinese alligators were collected at the Xuancheng Alligator Breeding Research Center. Eight alligator nests with significantly different hatching rates were selected, for a total of eighty eggs (*n* = 10 eggs/nest). The eggs were divided into two groups, with a hatching rate of 50% as the dividing line. Eggs with a hatching rate higher than 50% were assigned to the high-hatching-rate group, and those with a hatching rate lower than 50% were assigned to the low-hatching-rate group (high hatching rate = an average of 92.5% of eggs in the nests hatched, low hatching rate = an average of 39.1% of eggs in the nests hatched [27]). For sampling, three pieces of shell (approximately 1 cm^2^ each) were cut, including samples from the upper surfaces of the middle and both ends of each egg: the end where the young alligator naturally emerged from the shell was referred to as the cracked end (CE), the opposite end as the intact end (IE), and the middle as the middle section (ME). The eggshell membrane was removed, and pieces of shell were rinsed in distilled water and dehydrated in a graded series of alcohol. The outer surfaces of the pieces of shell were then sputter-coated with gold palladium (KYKY gold sprayer) and viewed at 20 kV under a scanning-electron microscope (SEM). Starting at an edge of each piece of shell at a magnification of 37×, open pores were counted on four consecutive SEM display screens along the middle of the piece of shell. Next, the pieces were oriented end-on, and micrographs (×100) were taken. Data from the three samples were pooled [28].

The average number of pores in the CE, ME, and IE regions was counted. When counting stomata on the eggshell surface, first, the stomatal numbers N_1_, N_2_, and N_3_ of the three microscopic images were obtained, and then the average stomatal number (N) was calculated.

We used MATLAB to measure the thickness of the cross-section of each eggshell and to calculate the mean. Using MATLAB (MathWorks Inc., USA), a GUI tool for measuring eggshell thickness from microscopic images of eggshell cross-sections was prepared. During the measurement, the length L_0_ of a 100-μm ruler in the image was measured by drawing a straight line on the microscopic image, and then the image widths L_1_, L_2_, and L_3_ at three different positions of the cross-section were measured by drawing a straight line. Subsequently, the average width (L) of the cross-section was calculated by the following formula.
L = (L_1_ + L_2_ + L_3_) / L_0_ × 100 / 3 (μm)

An appropriate amount of sample was weighed and placed into polytetrafluoroethylene liner of a high-pressure digestion tank. After addition of HNO, the sample was heated in a drying oven at 150 °C for 2 h. After cooling, the sample was transferred to a 25-mL volumetric flask and diluted to volume with ultrapure water as the test solution for metallic elements. A total of 25 mL of sample solution was accurately taken from the volumetric flask, placed into a 25-mL volumetric flask, diluted with ultrapure water to scale as a sample solution for testing metal elements, and directly tested for the content of each element by an AVIO200 (PerkinElmer, Shanghai, China) inductively coupled plasma emission spectrometer [29].

### 2.2. Data Analysis

Microsoft Excel was used to preprocess all data. First, the Kolmogorov–Smirnov test was performed to determine eggshell thickness and the number of pores in erosion craters. Next, a *t* test was performed to compare eggshell thicknesses between the groups, and the Mann–Whitney U test was used to compare the number of erosion-crater pores. Linear fitting was performed to determine the relationship between eggshell thickness and the number of erosion-crater pores. In addition, a Gaussian mixture model was used to cluster eggshell thicknesses and the number of pores in the CE, ME, and IE regions.

## 3. Results

### 3.1. Scanning Results for the Eggshells’ Outer Surfaces

The scanning-electron microscopy revealed that the outer surfaces of the Chinese alligator eggshells were covered with erosion craters and pores. A typical erosion crater spirals inwards and downwards, with a depression in the center. The erosion craters and pores on the eggshell surfaces were randomly distributed; while each pore was in the center of an erosion crater, not all the erosion craters contained pores (Figure 1a).

### 3.2. Effect of Alligator-Eggshell Thickness on the Hatching Rate

The eggshell thickness followed a normal distribution (*p* > 0.05; Appendix A). A *t* test was performed to compare the eggshell thicknesses between the two groups. The results showed that the eggshell thickness significantly differed between the high- and low-hatching-rate groups (*p* < 0.05; Appendix A, Table 1, Figure 1c,d).

The shells of the eggs with high hatching rates were significantly thicker than those of the eggs with low hatching rates (Figure 2a,c). The eggshell thickness was also positively correlated with the hatching rate (R = 0.912; Figure 3a).

Finally, we clustered the shell thicknesses of the CE, ME, and IE areas of the eggshells with a Gaussian mixture model. We found that eggshell thicknesses of 250–380 µm resulted in the highest hatching rates (Figure 4a,b, Appendix A).

### 3.3. Effects of the Number of Erosion-Crater Pores on the Surfaces of Alligator Eggs on Their Hatching Rates

The distribution of the number of erosion-crater pores on the surfaces of the alligator eggs was unknown (*p* < 0.05; Appendix A). The numbers of pores in the CE, ME, and IE regions were compared between the two hatching-rate groups using a Mann–Whitney U test. There were significant differences between the numbers of erosion-crater pores on the surfaces of the eggshells with different hatching rates (*p* < 0.05; Table 2, Figure 1a,b).

The results show progressive significance; the significance threshold was 0.05 and, thus, values lower than 0.05 indicated significance.

Specifically, the alligator eggs with high hatching rates had fewer erosion-crater pores on their surfaces in the CE, ME, and IE regions than the eggs with low hatching rates (Figure 2b,d). The number of erosion-crater pores was also negatively correlated with the hatching rate (R = −0.632; Figure 3b). Next, we compared the numbers of erosion-crater pores per unit area between the two hatching-rate groups. There was a significant difference (*p* < 0.01) between the numbers of pores per unit area on the surfaces of the eggshells with different hatching rates. Specifically, the eggs with high hatching rates had fewer pores per unit area on their surfaces than the eggs with low hatching rates (Figure 2b,d, Appendix A). Next, a Gaussian mixture model was used to cluster the number of erosion-crater pores per unit area; we found that the presence of 1–12 erosion-crater pores on the surfaces of the eggshells resulted in the highest hatching rates (Figure 4a,b, Appendix A). The number of erosion-crater pores was also negatively correlated with the eggshell thickness (R = −0.839; Figure 3c).

### 3.4. Eggshells’ Ca Contents Are Associated with the Hatching Rate

The variance in the Ca content in the two groups of samples with different hatching rates was homogeneous (*p* > 0.05). The Ca contents in the eggshells of the Chinese alligators were significantly correlated with the hatching rates (*p* < 0.05) and significantly differed between the hatching-rate groups (*p* = 0.025) (Table 3); specifically, the high-hatching-rate group had the shells with the highest Ca contents (Appendix A).

## 4. Discussion

### 4.1. Effect of Eggshell Thickness on the Hatching Rate

Eggshells that are excessively thin are at an increased risk of fracture, water evaporation, and bacterial penetration, which affect embryo development. However, eggshells that are overly thick have lower water contents during incubation and make it difficult for embryos to break during hatching, leading to embryo death. We found that eggshell thickness significantly affects the hatching rate of fertilized Chinese alligator eggs. The eggs with high hatching rates had significantly thicker shells than those with low hatching rates (*p* < 0.05). In addition, we found that eggshell thicknesses of 200–380 µm resulted in the highest hatching rates. Previous studies showed that the hatching rates of eggs with thick eggshells are significantly higher than those of eggs with thin eggshells, and that increased eggshell thickness increases the hatching rate of eggs, such as those of turkeys and geese [30,31]. Eggs with thin shells are more susceptible to bacterial infections and excessive dehydration [3]. Therefore, the eggshell thickness significantly affects the hatching rate. The eggshell thickness during incubation may vary between individuals [1]. Studies have shown that eggshell thickness decreases as females age; thus, eggshell thickness may indicate the age of the breeding individual, but this hypothesis needs to be further confirmed by future studies [1]. The eggshell thickness in Chinese alligators may similarly indicate the age of the female, but this hypothesis also needs further confirmation.

Eggshells vary in thickness across their surface. In the Chinese alligator, we found that ME is the thickest area of the eggshell, followed by CE, and that IE is the thinnest. Studies of eggshell thickness typically average the measurements of intact eggs at three locations: the eggshell tip, the middle circumference, and the blunt end [1]. However, in the present study, all the eggs had already hatched; thus, the end from which the alligator hatched was defined as the CE. The eggshell thickness of the CE was significantly greater than that of the IE. Moreover, maternal Ca is transferred directly to the shell during shell formation; in the eggshell, Ca is found in the form of calcium carbonate [12].

Eggshell thickness and egg weight are strongly correlated [32]. Eggshell thickness is difficult to measure; thus, future research would benefit from the determination of whether eggshell thickness can be estimated from egg weight. Moreover, future studies should determine the impact of egg weight on the hatching rates of Chinese alligators.

### 4.2. Effect of the Number of Erosion-Crater Pores on the Hatching Rate

As incubation progresses, the dense calcification layer of alligator eggs exhibits a gradual dissolution of crystals, producing concentric, stepped erosion craters. At the bottom of these erosion craters, the underlying honeycombed layer, which contains many gaps interconnected with other parts of the shell, is exposed [11]. The erosion is due to the byproducts of acid metabolism from bacteria in the nest material. This external degradation gradually increases the number of eggshell pores, reducing the integrity of the eggshell [11]. In the present study, the eggs with high hatching rates had significantly fewer erosion-crater pores on their surfaces than the eggs with low hatching rates. Thus, the eggs with low hatching rates probably experienced greater shell degradation and greater shell-weight loss, as well as exhibiting significantly more pores. Increases in the number of pores result in significantly greater gas and water exchange (leading to water loss) between the embryo and the external environment. Excessive water loss causes the embryo to become dehydrated, resulting in malformation; severe cases of dehydration result in embryo death, decreasing the hatching rate. Therefore, excessive pores on the eggshell surface are unfavorable for embryo development and the hatching of Chinese alligator eggs. Similarly, eggshell thickness and the number of pores significantly affect the development of crocodile embryos [33].

The number of pores is one of the most important characteristics of the shell structure. Nevertheless, both smaller and larger numbers of pores have negative impacts on embryo survival. The presence of excessively small numbers of pores leads to impaired oxygen exchange and, thus, increases embryo mortality [10], whereas overly high numbers of pores result in excessive gas and water exchange (and, thus, excessive water loss) between the embryo and the external environment, with severe negative impacts due to dehydration [9].

The eggshell cuticle is an uneven organic layer that covers the outer surface of the eggshell. The cuticle consists of an inner calcified layer and an outer noncalcified water-insoluble layer directly deposited on the calcified layer of the eggshell [34]. The deposition of the cuticle is highly important for preventing the invasion of microorganisms. Bacterial penetration is dependent on natural variation in the cuticle produced during deposition. Eggs with thick cuticles are not easily penetrated by bacteria [35], whereas the thinning of the eggshell cuticle causes embryo death (and, thus, a decreased hatching rate) due to bacterial penetration of the eggshell. Additionally, the cuticle serves as a barrier to prevent water loss and excessive dehydration of the egg material. Studies have shown that dehydration leads to a decline in the embryo’s innate immune function; moreover, the impact of dehydration on immune function may be greater than that on energy supply [36].

If the external humidity during incubation is excessively low, excessive water loss may occur, leading to the dehydration and death of the embryo. Gas exchange is also key to successful embryo development. During the gas-exchange process, external hypoxic conditions significantly affect the embryo’s yolk utilization and the size of the offspring [10]. Oxygen diffuses from the air into the embryo and is necessary for aerobic metabolism, while water vapor and carbon dioxide diffuse from the embryo into the air [37]. Gaseous oxygen must pass through the eggshell and the two shell membranes [38]. After gas exchange occurs, oxygen is transported to the tissues through the blood vessels and cardiovascular system [39]. If the hard eggshell surface is not corroded by bacterial byproducts in the nest material, the surface displays few pores, thus increasing the difficulty of meeting the gas-exchange requirements of the embryo. However, eggshells that are overly corroded and thinned by bacterial byproducts in the nest material may experience excessive water loss. Therefore, for alligator eggs with hard shells, moderate numbers of surface erosion-crater pores are conducive to gas exchange and the retention of water. However, an excessive number of erosion-crater pores thins the eggshell and leads to severe dehydration, which hampers the later growth and development of the embryo.

### 4.3. Effect of Eggshells’ Ca Content on the Hatching Rate

Detailed data on Ca metabolism in avian embryos are available, but there are few data regarding Ca metabolism in crocodilian embryos. The shell contributes substantially to embryonic Ca via transport across the chorioallantoic membrane in *Crocodylus novaeguineae* [40]. Clinical Ca deficiency has been reported in crocodiles fed red meat without supplementary Ca, as well as in animals housed under cover (which are thus unable to synthesize vitamin D_3_). Additionally, maternal Ca deficiency may translate into reduced Ca deposition in the eggshell, as suggested by Lance et al. (1983); however, no such effect has been formally or even anecdotally reported in crocodilians. In *Alligator mississippiensis*, most of the Ca utilized by the embryo originates in the shell, but the yolk may also serve as a temporary store of shell-derived Ca [41]. The present study showed that the Ca content of alligator eggshells has a significant effect on the hatching rate because the alligator eggs with high hatching rates had higher eggshell Ca contents than those with low hatching rates. The Ca in the eggshell is transported across the chorion to supply the embryo. Insufficient dietary Ca intake by the mother leads to a reduction in eggshell Ca deposition in crocodiles. Therefore, a substantial portion of the maternal Ca transferred to the embryo is provided through the eggshell.

The results of this study can inform future research. For example, the embryonic absorption of Ca from the shells of newly laid Chinese alligator eggs has not been determined. To assess the effect of maternal diet on eggshell Ca content and subsequent embryo viability, comparisons between captive and healthy wild populations should be conducted to identify shortcomings in the rearing of captive crocodiles.

## 5. Conclusions

The population status of the Chinese alligator, which is an endangered reptile species, is extremely worrisome. In the process of restoring the population of Chinese alligators, the hatching conditions of alligator eggs constitute an important factor. According to our research, the characteristic properties (thickness and erosion-crater pores) of alligator eggs are related to the egg quality and have a significant impact on the hatching rate. Additionally, the calcium content in the eggshell also has a positive correlation with the hatching rate. This project suggests that focusing on the hatching of Chinese alligator eggs will be an important direction for future conservation efforts.

## Figures and Tables

**Figure 1 animals-13-01405-f001:**
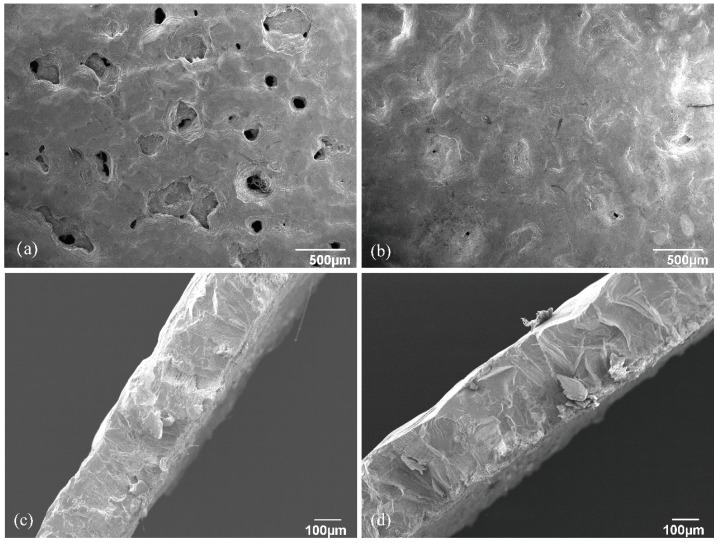
The outer surface of a Chinese alligator egg with erosion craters, some of which contain pores. The outer-shell surfaces of Chinese alligator eggs with low hatching rates (**a**) and high hatching rates (**b**), as well as the shell thickness of low-hatching-rate eggs (**c**) and high-hatching-rate eggs (**d**).

**Figure 2 animals-13-01405-f002:**
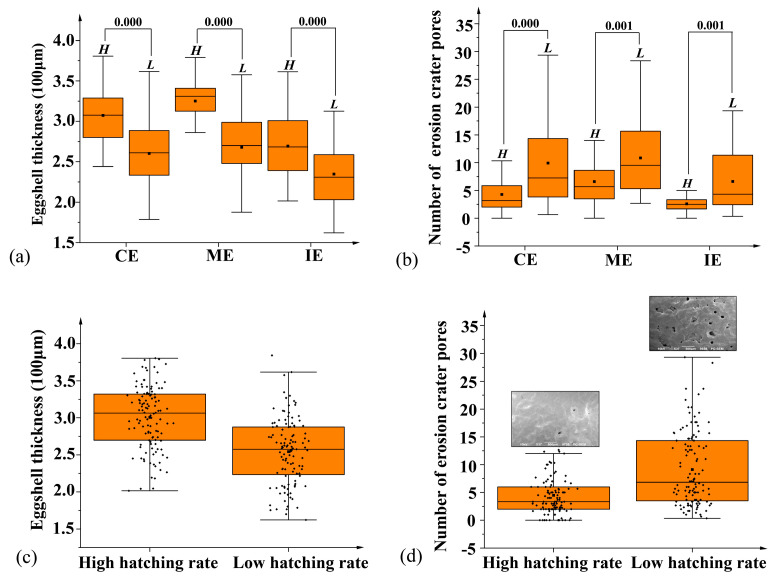
Box diagrams of the association between the hatching rate and eggshell thickness (**a**) and the number of erosion-crater pores (**b**) in the CE, ME, and IE regions, as well as the association of the hatching rate with the average number of pores (**d**) and average eggshell thickness (**c**).

**Figure 3 animals-13-01405-f003:**
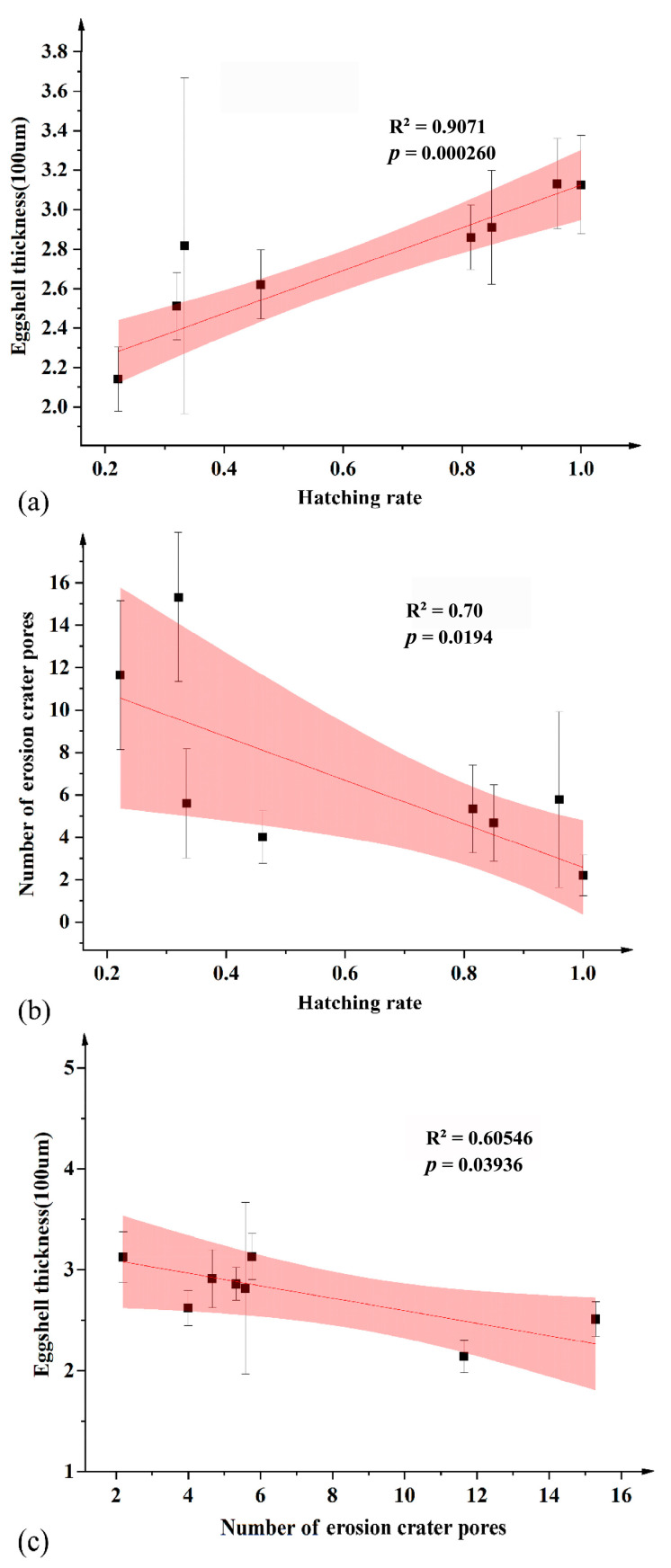
Eggshell thickness and hatching rate showed a significant positive correlation (*p* < 0.05) (**a**). The number of erosion-crater pores on the eggshell surfaces showed a significant negative correlation with the hatching rate (*p* < 0.05) (**b**). Eggshell thickness and the number of erosion-crater pores were significantly negatively correlated (*p* < 0.05) (**c**).

**Figure 4 animals-13-01405-f004:**
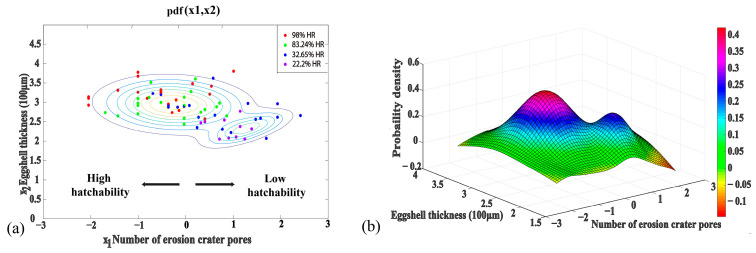
Gaussian mixture model used to cluster the eggs with similar hatching rates (HRs) according to the number of erosion-crater pores and the eggshell thicknesses (**a**,**b**) in the CE, ME, and IE regions.

**Table 1 animals-13-01405-t001:** Results of eggshell-thickness *t* tests.

	Levene Test			*T* Test of Mean Equation			
	F	Sig.	T	df	Sig. (Both Sides)	Mean Value Difference	Standard Deviation
CE	0.488	0.487	5.765	78	0.000	0.47	0.081
ME	4.825	0.031	68.369	78	0.000	0.571	0.084
IE	0.019	0.89	3.682	78	0.000	0.347	0.094
AVG	1.643	0.204	6.722	78	0.000	0.463	0.069

**Table 2 animals-13-01405-t002:** Mann–Whitney-U-test results for the number of erosion-crater pores on the surfaces.

	CE	ME	IE	AVG
Total N	80	80	80	80
Mann–Whitney U	1205.500	1141.500	1154.000	1222.500
Wilcoxon	2025.500	1961.500	1974.500	2042.500
Test statistic	1205.500	1141.500	1154.500	1222.500
Standard error	103.835	103.883	103.836	103.914
Standardized testing	3.905	3.287	3.414	4.066
Progressive significance (Z-sided test)	0.000	0.001	0.001	0.000

**Table 3 animals-13-01405-t003:** Results of *t* tests of the eggshells’ calcium contents.

Levene Test				*T* Test of Mean Equation		
F	Sig.	T	df	Sig. (Both Sides)	Mean Value Difference	Standard Deviation
0.02	0.889	2.569	12	0.025	47312.49917	18416.94162

## Data Availability

The datasets used and analyzed during the current study are available from the first author upon reasonable request.

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
