# Peer review of "Effects of Eggshell Thickness, Calcium Content, and Number of Pores in Erosion Craters on Hatching Rate of Chinese Alligator Eggs"

_animals, 2023, doi:10.3390/ani13081405_

Round 1
Reviewer 1 Report
I think this is valuable information although I would have liked to see some recommendations regarding adult animal health, as croc farming and ranching is a significant conservation tool and this is highly relevant to these operations.
Author Response
-
Dear Reviewer,
Thank you for your advice. We agree with you very much. Adult animal health has an important effect on the hatching rate of Chinese alligator eggs. However, due to the difficulty in measuring the physical condition of adult crocodiles in the wild, this part is missing. In the future, we will make new integration of this part based on new technologies and methods according to your suggestions.
Please do not hesitate to contact us with any questions you may have. We’re looking forward to further news from you.
Yours sincerely
Tao pan

Reviewer 2 Report
The manuscrips is devoted to the important point of reproductive biology of the Chinese alligator (Alligator sinensis), found only in a small region in southeastern Anhui Province and listed as critically endangered (CR) by the International Union for Conservation of Nature (IUCN) due to its current population trend. The authors show interesting results for clarification of the problem of effects of eggshell parameters on hatching rate. However it will be useful to include the chapter "Conclusion" to have the short summary of results and their interpretation.
Author Response
Dear Reviewer,
Thanks. We had added the conclusion part into the revised manuscript following your suggestion.
Conclusions: As an endangered reptile species, the population status of the Chinese alligator is extremely worrisome. In the process of restoring the population of Chinese alligators, the hatching condition of alligator eggs is an important factor. In our research, the characteristic properties (thickness and corrosion pits) of alligator eggs, are related to the egg quality and have a significant impact on the hatching rate. Additionally, the calcium content in the eggshell also has a positive correlation with hatching. This project suggests that focusing on the hatching of Chinese alligator eggs will be an important direction for future conservation efforts.
Please do not hesitate to contact us with any questions you may have. We’re looking forward to further news from you.
Yours sincerely
Tao pan

Reviewer 3 Report
I appreciate the work done, and its importance for contributing to our understanding on the reproductive biology of this interesting taxon.
There are places where additional English editing will be required (notably in paragraph 1 of the Materials and Methods when you discuss the areas of shell sampled and their abbreviations (something appears to be missing -- "where the young alligator naturally emerged from the shell as the cracked end (LKD), the opposite end as the intact end" -- this does not translate well in English -- Perhaps "We used the abbreviation LKD for the cracked end of the egg where the young alligator naturally emerged from the shell, the opposite, intact, end as WZD..."; see also paragraph four of Materials and Methods (starting with "Weight a proper amount of sample..." and continuing through to just above section 2.2; See also the Ethics statement, which lacks proper English syntax).
A few suggested changes:
P. 2: "Once hardened (opaque)... should be changed to "Once hardened (i.e., when they become opaque..." At the end of that sentence, "contents" should be "content."
The abbreviations LKD, WZD, and ZJD make no sense (at least in English), so I had to continually go back and refresh my memory of what they meant. Alternative abbreviations might be helpful.
Table 1: "Standard deviarian" should be "Standard deviation"
Just below Table 1 : "...had fewwer erorion crater pores on the surface at the LKD, ZJD, and WZD that eggs..." should be "... THAN eggs" (spell-checker will not catch that).
The axes labels on Figures 2 and 3 are too small to read.
In the discussion, when you mention turkeys and geese, provide citation(s) to that work. Just below there, you use the word ":theory" but it is not a theory, it is a hypothesis. Adjust accordingly. At end of that paragraph, your syntax implies that you have confirmed that thickness ay indicate age of the female ("further confirmed"), but you have not so alter the wording accordingly.
At the bottom of page 11 you mention "...some eggs had already hatched; thus...The LKD had a significantly thicker eggshell than the WZD." For this, please clarify whether you made this assessment only eggs that had already hatched, or also on eggs that had not hatched, and you broke open (in which case, how did you determine LKD vs WZD??
On top of page 12 you mention interspecific comparisons, but it not clear to me that you made comparisons with other species. please clarify this.
P. 12, section 4.2, 7 lines down: "...crater pores on the surface that eggs..." should be "THAN eggs" (again, a spell check with not catch this).
On the next line, "...greater shell weight loss" might be better as "greater weight loss of the shell"
P. 13, 8 lines down "...the thinning of the eggshell cuticle causes of embryo death" --- delete the "of". Two lines later "...water loss and preventing excessive..." should be "...water loss and to prevent excessive"
P. 13, 6 lines above section 4.3: "...surface will have few pores, and increasing the difficulty" change "and" to "thus"
P. 13, two lines above section 4.3: "...pores thins the eggshell leads..." should be "...pores thins the eggshell and leads..."
top of page 14: "There is very little data" should be "there are few data"
p. 14, 7 lines down: "...or even anecdotally reported in crocodiles" should be "...or even anecdotally reported in crocodilians"
three lines lower, change "...as the alligator eggs with a high..." should be "...because the alligator eggs with a high..."
At end of the acknowledgements, you list taxon names -- all should e italicized (I don't know what Odorax aurea or Ichthyosaurus Banna are).
In the References section, there are numerous citations that include scientific names, but those are names are not italicized. Any of these that were originally published in Chinese should be rendered with BOTH the original Chinese titles and an English translation. IT is difficult to find Chinese literature if only a translation is provided.
Author Response
Dear Reviewer,
Thank you for your valuable suggestions. The following are our modifications and replies:
1、There are places where additional English editing will be required (notably in paragraph 1 of the Materials and Methods when you discuss the areas of shell sampled and their abbreviations (something appears to be missing -- "where the young alligator naturally emerged from the shell as the cracked end (LKD), the opposite end as the intact end" -- this does not translate well in English -- Perhaps "We used the abbreviation LKD for the cracked end of the egg where the young alligator naturally emerged from the shell, the opposite, intact, end as WZD..."; see also paragraph four of Materials and Methods (starting with "Weight a proper amount of sample..." and continuing through to just above section 2.2; See also the Ethics statement, which lacks proper English syntax).
- Answer: Thank you! We have revised this sentence. The revised sentence is: For sampling, three pieces of shell (approximately 1 cm2 each) were cut, including samples from the upper surface of the middle and both ends of each egg: the end where the young alligator naturally emerged from the shell was referred to as the cracked end (CE), the opposite end as the intact end (IE), and the middle as the middle section (ME). And checked and modified the grammar of paragraph 4 and other paragraphs in Materials and Methods according to the reviewer's opinion. Modified statement: A proper amount of sample was weighed and placed into a polytetrafluoroethylene liner of a high-pressure digestion tank. HNO was added, and then the sample was heated in a drying oven at 150 °C for 2 h. After cooling, the sample was transferred to a 25 mL volumetric flask and diluted to volume with ultrapure water as the test solution for metallic elements. A total of 25 mL of sample solution was accurately taken from the volumetric flask, placed into a 25 mL volumetric flask, diluted with ultrapure water to scale as a sample solution for testing metal elements, and directly tested for the content of each element by an AVIO200 inductively coupled plasma emission spectrometer.
2、P. 2: "Once hardened (opaque)... should be changed to "Once hardened (i.e., when they become opaque..." At the end of that sentence, "contents" should be "content."
- Answer: Thank you! I have changed "contents" to "content." according to the reviewer's opinion, please check.
3、The abbreviations LKD, WZD, and ZJD make no sense (at least in English), so I had to continually go back and refresh my memory of what they meant. Alternative abbreviations might be helpful.
- Answer: Thank you! The abbreviations LKD, WZD and ZJD have been changed to CE, IE and ME according to the reviewer's comments.
4、Table 1: "Standard deviarian" should be "Standard deviation".
- Answer: Thank you! "Standard deviarian" has been changed to "Standard
deviation" according to the reviewer's opinion.
5、Just below Table 1 : "...had fewwer erorion crater pores on the surface at the LKD, ZJD, and WZD that eggs..." should be "... THAN eggs"
- Answer: Thank you! That has been changed to than, please check.
6、The axes labels on Figures 2 and 3 are too small to read.
- Answer: Thank you! The axis labels in Figures 2 and 3 have been enlarged,
please check.
7、In the discussion, when you mention turkeys and geese, provide citation(s) to that work. Just below there, you use the word ":theory" but it is not a theory, it is a hypothesis. Adjust accordingly. At end of that paragraph, your syntax implies that you have confirmed that thickness ay indicate age of the female ("further confirmed"), but you have not so alter the wording accordingly.
- Answer: Thank you! The Turkey and goose literature has been inserted and the word "theory" has been replaced with "hypothesis". Please check.
8、At the bottom of page 11 you mention "...some eggs had already hatched; thus...The LKD had a significantly thicker eggshell than the WZD." For this, please clarify whether you made this assessment only eggs that had already hatched, or also on eggs that had not hatched, and you broke open (in which case, how did you determine LKD vs WZD?
- Answer: Thank you! We made this assessment only eggs that had already hatched, not the unhatched eggs. We defined the end of the hatchlings that naturally emerged as the cracked end, and the complete end as the intact end. And we have modified the sentence, please check.
9、On top of page 12 you mention interspecific comparisons, but it not clear to me that you made comparisons with other species. please clarify this.
- Answer: Thank you! Our study has not involved interspecific, after our consideration, the sentence has been adjusted. We have revised this sentence, please check. The part can be changed to eggshell thickness and egg weight are strongly correlated. Eggshell thickness is difficult to measure; thus, future research would benefit from determination of whether eggshell thickness can be estimated from egg weight. Moreover, future studies should determine the impact of egg weight on the hatching rate of Chinese alligators.
10、P. 12, section 4.2, 7 lines down: "...crater pores on the surface that eggs..." should be "THAN eggs" (again, a spell check with not catch this).
- Answer: Thank you! We have changed the "that" to "than" here, please check.
11、On the next line, "...greater shell weight loss" might be better as "greater weight loss of the shell".
- Answer: Thank you! This sentence has been modified according to the
reviewer's opinion.
12、P. 13, 8 lines down "...the thinning of the eggshell cuticle causes of embryo death" --- delete the "of". Two lines later "...water loss and preventing excessive..." should be "...water loss and to prevent excessive"
- Answer: Thank you! The superfluous word of has been deleted here and
preventing has been replaced with prevent.
13、P. 13, 6 lines above section 4.3: "...surface will have few pores, and increasing the difficulty" change "and" to "thus".
- Answer: Thank you! We have changed "and" here to "thus",please check.
14、P. 13, two lines above section 4.3: "...pores thins the eggshell leads..." should be "...pores thins the eggshell and leads..."
- Answer: Thank you! We have made the modification, please check.
15、top of page 14: "There is very little data" should be "there are few data"
- Answer: Thank you! We have made the modification, please check.
16、p. 14, 7 lines down: "...or even anecdotally reported in crocodiles" should be "...or even anecdotally reported in crocodilians".
- Answer: Thank you! We have made the modification, please check.
17、three lines lower, change "...as the alligator eggs with a high..." should be "...because the alligator eggs with a high..."
- Answer: Thank you! We have made the modification, please check.
18、At end of the acknowledgements, you list taxon names -- all should e italicized (I don't know what Odorax aurea or Ichthyosaurus Banna are).
- Answer: Thank you! We have changed Odorax aurea and Ichthyosaurus Banna to Odorrana tormota and Ichthyophis bannanicus, please check.
19、In the References section, there are numerous citations that include scientific names, but those are names are not italicized.
- Answer: Thank you! We have changed the scientific names quoted in the literature to italics
20、Any of these that were originally published in Chinese should be rendered with BOTH the original Chinese titles and an English translation. IT is difficult to find Chinese literature if only a translation is provided.
- Answer: Thank you for your advice. We have reviewed this part of content according to your suggestion. I feel that there are misreferences in the application of this paper, so it may be more appropriate to replace it with a new one. In the manuscript, changes have been made accordingly.
Reference:
- Ferguson, M. W. The reproductive biology and embryology of the crocodilians. Biology of the Reptilia. 1981, 12, 330-491.
- Packard, M. J.; Packard, G. C. Mobilization of calcium, phosphorus, and magnesium by embryonic alligators (Alligator mississippiensis). AM J PHYSIOL-REG I. 1989, 257 (6), R1541-R1547.
- Tang, W.; Zhao, B.; Chen, Y.; Du, W. Reduced egg shell permeability affects embryonic development and hatchling traits in Lycodon rufozonatum and Pelodiscus sinensis. Integrative Zoology. 2018, 13 (1), 58-69.
- Brown, G. J.; Forbes, P. B.; Myburgh, J. G.; Nöthling, J. O. Calcium and phosphorus in unbanded eggs of the Nile crocodile (Crocodylus niloticus). Res. 2020, 51 (8), 3403-3411.
Please do not hesitate to contact us with any questions you may have. We’re looking forward to further news from you.
Yours sincerely
Tao pan

Reviewer 4 Report
Reviewer's report
Date: March 13, 2023
Journal: Animals
Manuscript ID: animals-2258030
Type of manuscript: Article
Title: Effects of eggshell parameters (eggshell thickness, calcium content,
and number of pores in erosion craters) on hatching rate of Chinese alligator
eggs
Authors: Naijing Zhang, Huabin Zhang, Guangwei Fan, Ke Sun, Qingqing Jiang,
Zhuowen Lv, Boyang Han, Zhenyuan Nie, Yujie Shao, Baowei Zhang, Yongkang
Zhou, Tao Pan *, Xiaobing Wu *
The authors have investigated the relationship between eggshell parameters and hatching rate in critically endangered Chinese alligator (Alligator sinensis). The results are important for the conservation biology of this species. The results of the manuscript may provide useful cues to scientists interested in conservation biology. However, the manuscript is messy. It contains numerous inaccuracies. At the present condition the manuscript is not appropriate for the journal Animals.
Major points
1. The text of the manuscript is not properly formatted.
2. The English language is frequently not enough clear.
3. Figures should be ordered as they are mentioned in the text.
Minor points
Page 1: Change the title of the manuscript from “Effects of eggshell parameters (eggshell thickness, calcium content, and number of pores in erosion craters) on hatching rate of Chinese alligator eggs” to “Effects of eggshell thickness, calcium content, and number of pores in erosion craters on hatching rate of Chinese alligator eggs”.
Page 1, Abstract, last sentence is not completed. Change: “Furthermore, our findings can inform future studies.” To “Furthermore, our findings can inform future studies, which are vitally necessary to survive critically endangered Chinese alligator species.”
Page 3: The first formula is not necessary. It is an arithmetic average.
Page 3: Change um to μm.
Page 3 last paragraph and page 4 first paragraph: Consider English revision.
Page 5, Table 1: Format the table.
Page 5: The text should mention the images in order: a, b, c, and d.
Page 5: The probability symbol (P) must be in Italics.
Page 7: The comments for the Table 2 must be corrected.
Page 8m Table 3: The Levene test is not significant, which should be mentioned in the text.
Page 8, Figures 2 and 3: The symbols in the figures should be enlarged.
Page 12, paragraph 4.2: Consider English revision for the second sentence.
Page 14: Last three sentences should be in a separate paragraph.
Author Response
Dear Reviewer,
Thank you for your valuable suggestions. The following are our modifications and replies:
1、Page 1: Change the title of the manuscript from “Effects of eggshell parameters (eggshell thickness, calcium content, and number of pores in erosion craters) on hatching rate of Chinese alligator eggs” to “Effects of eggshell thickness, calcium content, and number of pores in erosion craters on hatching rate of Chinese alligator eggs”.
- Answer: Thank you! We have modified the title according to the reviewer's opinion, please check.
2、Page 1, Abstract, last sentence is not completed. Change: “Furthermore, our findings can inform future studies.” To “Furthermore, our findings can inform future studies, which are vitally necessary to survive critically endangered Chinese alligator species.”
- Answer: Thank you! This sentence has been modified, please check.
3、Page 3: The first formula is not necessary. It is an arithmetic average.
- Answer: Thank you! This formula has been deleted, and this place has been adjusted, please check.
4、Page 3: Change um to μm.
- Answer: Thank you! We have changed um to μm, please check.
5、Page 3 last paragraph and page 4 first paragraph: Consider English revision.
- Answer: Thank you! The English of this section has been revised. This part is modified to a proper amount of sample was weighed and placed into a polytetrafluoroethylene liner of a high-pressure digestion tank. HNO was added, and then the sample was heated in a drying oven at 150 °C for 2 h. After cooling, the sample was transferred to a 25 mL volumetric flask and diluted to volume with ultrapure water as the test solution for metallic elements. A total of 25 mL of sample solution was accurately taken from the volumetric flask, placed into a 25 mL volumetric flask, diluted with ultrapure water to scale as a sample solution for testing metal elements, and directly tested for the content of each element by an AVIO200 inductively coupled plasma emission spectrometer.
6、Page 5, Table 1: Format the table.
- Answer: Thank you! Table formats have been modified, please check.
7、Page 5: The text should mention the images in order: a, b, c, and d.
- Answer: Thank you! The order of the pictures has been adjusted, please check.
8、Page 5: The probability symbol (P) must be in Italics.
- Answer: Thank you! All P in the article have been changed to P,please check.
9、Page 7: The comments for the Table 2 must be corrected.
- Answer: Thank you! The comments in Table 2 have been corrected. The revised comment is the results show progressive significance; the significance threshold is 0.05, and thus, values less than 0.05 indicate significance.
10、Page 8m Table 3: The Levene test is not significant, which should be mentioned in the text.
- Answer: Thank you! This place has been adjusted, please check.
11、Page 8, Figures 2 and 3: The symbols in the figures should be enlarged.
- Answer: Thank you! The symbols in Figures 2 and 3 have been enlarged, please check.
12、Page 12, paragraph 4.2: Consider English revision for the second sentence.
- Answer: Thank you! This sentence has been amended in English. The revised sentence is at the bottom of these erosion craters, the underlying honeycombed layer, which contains may gaps interconnected with other spots in the shell, is exposed.
13、Page 14: Last three sentences should be in a separate paragraph.
- Answer: Thank you! The place has been adjusted, please check.
Please do not hesitate to contact us with any questions you may have. We’re looking forward to further news from you.
Yours sincerely
Tao pan

Round 2
Reviewer 4 Report
The manuscript was significantly improved. It is appropriate now for the journal Animals.
Author Response
Dear Reviewer,
Thank you for your approval of our revised manuscript and thank you again.
Please do not hesitate to contact us with any questions you may have. We’re looking forward to further news from you.
Yours sincerely
Tao pan
